# Broadening the Veterinary Consultation: Dog Owners Want to Talk about More than Physical Health

**DOI:** 10.3390/ani13030392

**Published:** 2023-01-24

**Authors:** Helena Hale, Emily Blackwell, Claire Roberts, Emma Roe, Siobhan Mullan

**Affiliations:** 1Bristol Veterinary School, University of Bristol, Langford BS40 5DU, UK; 2School of Geography and Environmental Sciences, University of Southampton, Southampton SO17 1BJ, UK; 3School of Veterinary Medicine, University College Dublin, D04 V1W8 Dublin, Ireland

**Keywords:** dog, quality of life, welfare, veterinary, consultation

## Abstract

**Simple Summary:**

Formal tools are available to aid veterinary assessment of canine quality of life. However, they are rarely applied in practice, and previous research suggests a veterinary perception of owner resistance to their use. Through an online questionnaire, we found that almost all UK dog owners (95.8%) were comfortable discussing their dogs’ quality of life with their vets, yet only a third of owners (32%) reported this topic to have been raised by their vets. Furthermore, the majority of owners (70.8%) were interested in accessing tools to assess their dog’s quality of life, but very few had experienced any form of formal health or well-being assessment tool (4.4%) with their vets. A subset of owners was interviewed about their experiences with such tools, and three main themes were generated from their feedback: ‘Use of assessment tools supports client-vet relationship and empowers owners’, ‘owners want to talk about holistic dog care’, and ‘owner feelings on the wider application of assessment tools’. Overarching findings suggest that owners want to discuss QOL and are interested in using formal assessment tools. Indeed, the uptake of tools appears to be valuable in improving the vet-client relationship and owner confidence in the treatment of their dogs.

**Abstract:**

Few veterinary professionals use formal quality of life (QOL) assessment tools despite their recommendation from veterinary governing bodies to enable holistic welfare assessments and target welfare improvement strategies. Perceived barriers include resistance from owners, and this study aimed to elucidate understanding of dog owner engagement with conversations and tools relating to QOL. An online survey that investigated owner experience, comfort, and opinions about vet-client discussions on topics connected to canine health and well-being, including QOL, was completed by 410 owners. Almost all owners (95.8%) were reportedly comfortable discussing QOL, yet only 32% reported their vets had addressed it. A high proportion of owners (70.8%) expressed interest in assessment tools, but only 4.4% had experienced one, none of which were QOL tools per se. Semi-structured interviews of a sub-set of four owners provided a more in-depth examination of their experience of a health and well-being assessment tool. Thematic analysis generated three themes: ‘Use of assessment tools supports client-vet relationship and empowers owners’, ‘Owners want to talk about holistic dog care’, and ‘Owner feelings on the wider application of assessment tools’. Overall, our findings suggest that owners want to broaden the veterinary consultation conversation to discuss QOL and are interested in using tools, and therefore veterinary perceptions of owner-related barriers to tool application appear unfounded. Indeed, tool uptake appears to improve the vet-client relationship and boost owner confidence.

## 1. Introduction

Recent estimates report that 90% of dogs in the UK (United Kingdom) are registered with a veterinary practice [1]. Veterinary professionals, therefore, have the potential to play a significant role in improving and maintaining the welfare of most companion dogs. There is a growth in interest in what can be achieved to improve animal welfare through the style of conversation a veterinary professional can have with those caring for animals [2,3]. It has been demonstrated that the style of the conversation can shape how farmers respond to veterinary advice and that this is achieved through shifting away from a paternalistic framing of the conversation towards a mutualistic, relationship-centred conversation style.

There appears to be a gap in communication and expectations between veterinary professionals and clients in companion animal practice, for example, in euthanasia [4] and preventative medicine [5] consultations. One potential method for bridging these gaps is the use of quality of life (QOL) assessment tools, which have facilitated patient-physician communication in human medicine [6]. In companion animal medicine, significantly more clients using a QOL assessment tool as part of a routine consultation were reported as feeling that all their questions about their dog’s care had been answered, compared to those not using a tool [7] and 81% of clients felt more involved in their animal’s care after completing a QOL assessment [8].

Indeed, the use of assessment tools to measure quality of life (QOL) has been recommended in veterinary practice [9,10], and numerous canine QOL assessment tools exist in both the peer-reviewed [11] and non-peer-reviewed literature. These tools have been developed often with the expectation that they will be of value and useable within the conversation space between veterinary professionals and dog owners. However, there appears to be a low uptake of these tools in practice, with veterinary professionals instead preferring to rely on a combination of clinical experience and intuition to assess QOL in dogs [12].

The existing literature which has explored veterinary professionals’ experience suggests that one of the main barriers to using canine QOL assessment tools was that they felt owners would be resistant to the idea [12]. This mirrors the discussion of preventive health care in companion animal veterinary practice, where veterinary surgeons thought that owners would not be interested in ‘extras’ in their consultations and would only want to talk about the specific reason for their visit [13]. However, evidence would suggest that these views of veterinary professionals on companion dog owners may be incorrect. Owners discussing preventive healthcare for their dogs reported that they would like to have longer and more thorough consultations with their veterinary surgeon [13]. Additionally, a recent trial of one general QOL assessment tool in a companion animal veterinary practice reported the majority (83.7%) of owners who completed the assessment before a check-up appointment thought it to be valuable, and most reported that they would be interested in completing one before every routine consultation [7].

The current study used a mixed methods approach to investigate dog owners’ experiences and attitudes towards discussing various topics relating to canine health and well-being with a veterinary professional and whether they had encountered the use of a formal assessment tool—and if so—how this experience was received. A survey was used to gain knowledge on owner-vet conversation topics, and semi-structured interviews were applied to deepen understanding of owner experiences and perceptions of formal assessment tools. The overall aim was to investigate whether the veterinary perception that owners may be resistant to using formal assessment tools or discussing topics beyond physical health is well-founded.

## 2. Materials and Methods

### 2.1. Online Survey

An online survey entitled “UK dog owner survey: conversations at the vets” was created in the Jisc Online Survey platform and, following piloting, was made available with no changes between 17th May to 28th June 2021 (see Appendix A). It consisted of 20 questions, took 5–10 min to complete and was advertised on social media (focused on dog-related online Facebook fora and Twitter). Participants were eligible if they were UK dog owners, aged over 18, and gave consent for their data to be used in the study. Data were collected anonymously, although contact details were provided by owners who were happy to be considered for further research.

Demographic data were collected on age (18–24, 25–34, 35–44, 45–54, 55–64, and 65+ years), gender (female, male, and prefer not to say), number of dogs currently owned, the length of time they had owned their current dog(s), and whether they had owned or co-owned a dog previously. Owners were also asked about the number of annual vet visits made and whether they worked in a dog or animal-related profession (vet or vet nurse, animal behaviourist, dog trainer, dog walker, animal carer, researcher of animal behaviour, other animal-related professions, and none of the above).

The next section asked about 17 different topics relating to the dogs’ physical and emotional health and well-being: puppy socialisation or training, creating a safe and comfortable environment, e.g., sleeping areas, travel, hazards (poisonous foods and plants), body weight, diet/nutrition, neutering, behaviour problems, fear-related issues, happiness, general welfare, quality of life, specific health problems, pain management, socialising with other dogs/people, mental stimulation, preventative medicine, senior health, and euthanasia.

Respondents were asked whether a vet or vet nurse had ever discussed each topic with them and how comfortable owners felt/would feel discussing each topic (very comfortable, quite comfortable, neither comfortable nor uncomfortable, quite uncomfortable, or very uncomfortable). Owners were also asked when (should be routinely discussed in each consult, should be discussed only if owner requests, should be discussed if vet or vet nurse thinks it’s relevant, not appropriate for vet or vet nurse to discuss, unsure, and other) and how (within a routine 10–15 min consultation, within a separate 10–15 min consultation, not appropriate for vet or vet nurse to discuss, unsure, and other) they felt each topic should be addressed by a vet or vet nurse, and to choose three that they would most wish to discuss in relation to their dog’s current life.

Owners were further asked if their vet or vet nurse had ever used a paper or computerised questionnaire to guide a conversation with them about their dog’s health or well-being or requested them to complete one themselves (yes or no)—and if yes—to state the purpose (free text) and how useful they found it (extremely useful; quite useful; unsure; quite un-useful; extremely un-useful). They were also asked whether they had ever used an App designed for this purpose and provided with an example (PetDialog™(Zoetis, Leatherhead, UK)) [14], Davies et al., 2020). If they answered yes to any of these, they were asked to state the purpose and how useful they found this. Finally, owners were asked if they would want to access tools designed to assess different aspects of their dog’s behaviour, health or well-being if they were widely available (yes, no, or unsure). Respondents were then given the opportunity to provide their email addresses if they were willing to be contacted for further research.

Data were extracted from Jisc Online Surveys and coded in Excel for non-parametric analysis using IBM SPSS Statistics 26. Chi-square tests were performed to investigate relationships between variables, and Fisher’s Exact test statistics were used where >20% of cell counts were less than 5.

### 2.2. Semi-Structured Interviews—Design and Analysis

Survey participants who had completed a questionnaire about their dog’s health or well-being with (or for) their vets and had given consent to be contacted for participation in additional related research were emailed to request an interview. They were provided with participation information, and consent was obtained before an appointment date was arranged.

The survey results, along with the findings from a previous study focusing on vet perception of the application of Quality of Life tools with dog owners [12], informed the development of the semi-structured interview questions around key topics: owner experience of discussing quality-of-life with their vet and their understanding of it, the type of health/well-being questionnaire, whether the experience led to any changes in care, treatment or perceptions, whether engaging with the questionnaire impacted owners emotionally, and thoughts on the future use and wider application of such tools. The interview was pilot tested, and no changes were made. Interviews were conducted by a single researcher (HH) over Zoom (Zoom Communications (UK) PLC, London, UK.) with the primary caregiver who had completed the survey and recorded. To maintain anonymity, interviewees were given a unique ID number to be used during the Zoom call and in data handling (and for the purpose of this paper, pseudonyms for dog and owner were used). The four audio recordings were transcribed by the same single researcher.

Transcripts were imported into NVivo (Release 1.6.1, QSR International UK Ltd, Warrington, UK) for coding. A reflexive, inductive approach was taken to code the data drawn from the semi-structured interviews and for the subsequent development of analytical themes. In other words, the themes were actively generated by the researcher’s interpretation of the data as opposed to being pre-set. However, the analysis was partially deductively informed due to the pre-set questions of the semi-structured interviews [15,16].

## 3. Results

### 3.1. Survey Demographics

Demographic information for survey participants is reported in Table 1. In total, 410 owners completed the survey, of which 380 were female (93%), and 29 were male (7%). There was a relatively even spread of age categories, and 24% worked in an animal-related profession. The majority of respondents (64%) had one dog at the time of survey completion, 26% had two, and 9.6% had three or more, with 63.9% of participants having owned or co-owned a dog before their current dog(s). There was a range in length of ownership of the current dog, with the most common category being 3–5 years (26.8%).

Owners reported, on average, attending an appointment with their dog with a vet or a vet nurse most commonly twice a year (32.5%), followed by 3–4 times a year (25.9%), once a year (24.4%), more than five times a year (11%), and 6.1% less than once a year. It was considered that owning more than one dog might impact the total number of annual vet visits made, but responses for the 36% of participants with a single dog revealed a similar pattern, where owners most commonly visited twice a year (35.6%), followed by 3–4 times a year (26.5%), once a year (25%), more than five times a year (8%), and less than once a year (4.9%).

### 3.2. Discussions at the Vets

Owners were asked whether their vet or vet nurse had ever discussed each of the 17 health and welfare topics about their current dog(s; Table 2). The most discussed topics reported by owners were preventative medicine (90.7%), body weight (80.1%), specific health problems (76.0%), neutering (70.9%), diet/nutrition (61.7%), and general welfare (54.5%). Fewer than 20% of owners reported discussing fear-related issues (19.8%), socialising with other dogs/ people (19.3%), euthanasia (18.0%), creating a safe and comfortable environment (15.2%), and mental stimulation (11.0%).

For all topics, the majority of owners reported that they would be very comfortable discussing them, the top three being specific health problems, preventative medicine, and pain management (Table 3). The topic that owners were least comfortable about was euthanasia. Where fewer than 70% of owners reported being very comfortable, the topics all related to behaviour and emotional well-being (and euthanasia). Fisher’s Exact tests revealed no significant associations between owner age category or gender and comfort discussing each topic (*p* > 0.05 for all).

For 10 of the topics (58.8%), those not in animal-related roles were significantly more likely to report being very or quite comfortable discussing the topics than those in animal-related roles (puppy socialisation or training, *p* = 0.015; creating a safe and comfortable environment, *p* = 0.016; diet and nutrition, *p* < 0.001; behaviour problems, *p* < 0.001; fear related issues, *p* = 0.004; happiness, *p* = 0.011; general welfare, *p* = 0.049; socialising with other dogs/people, *p* < 0.001; mental stimulation, *p* < 0.001; preventative medicine *p* = 0.043).

For 11 topics (64.7%), the majority of owners thought that they should be discussed ‘if the vet or vet nurse thinks it’s relevant’. These were: puppy socialisation or training (62.3%), creating a safe and comfortable environment (59.5%), neutering (63.6%), behaviour problems (51.6%), fear-related issues (53.8%), specific health problems (46.1%), pain management (59.4%), socialising with other dogs/people (52.1%), mental stimulation (47.1%), senior health (64.6%), and euthanasia (71.9%). The remaining topics all had a majority of owners that thought they should be addressed ‘routinely in every consult’: body weight (62.4%), diet/nutrition (48%), happiness (43%), general welfare (60.6%), preventative medicine (60.7%), and quality of life (51.4%). Very few owners felt that it was ‘not appropriate for the vet or vet nurse to discuss’ topics; topics relating to behaviour and emotional welfare showed the highest number of owners who felt this way: behaviour problems (4.2%), fear-related issues (3.4%), socialising with other dogs/people (3.7%), puppy socialisation and training (3.2%), and mental stimulation (3.7%). The topics with the highest number of owners who selected that it should only be discussed ‘if the owner requests it’ were behaviour problems (25.7%), fear-related issues (26%), socialising with other dogs/people (17.6%), mental stimulation (16.4%) and neutering (18.7%; Figure 1).

For 14 topics, a majority of respondents thought they should be addressed ‘within a routine 10–15-min consultation’. For the remaining three topics (behaviour problems, fear-related issues, and euthanasia), the majority felt they should be discussed ‘within a separate 10–15-min consultation’. Once again, very few people felt it was ‘not appropriate for a vet or vet nurse to discuss’ the various topics in the context of a veterinary consultation (maximum 6.4% for socialising your dog with other dogs and/or people; Figure 2).

Owners were asked to select three aspects of their dog’s current life that they would like to discuss with their vet or vet nurse (Figure 3); the most popular topics were a specific health problem (36.7%), general welfare (35.2%), and preventative medicine (28.1%), with fewer than 4% selecting euthanasia.

Thirty-nine owners (9.5%) suggested additional topics to discuss with their vet relating to health (dental care or disease, allergies, lumps, breed-related issues, first aid, preventative health care), medication, pain detection, alternative/complementary therapies, physiotherapy, recommendations for local dog businesses, general dog behaviour and training, behavioural issues (aggression and anxiety), sourcing and rehoming dogs breeding, insurance and foreign travel requirements.

### 3.3. Use of Assessment Tools

Eighteen (4.4%) owners reported that their vet had used a paper or computerised questionnaire to ask about their dog’s health or well-being or invited them to complete one, whilst 32 (7.8%) were unsure, and 360 (87.8%) had not experienced this. There were varying reports about the usefulness of the questionnaire, where most thought it was ‘quite useful’ (n = 9), followed by ‘extremely useful’ (n = 3), ‘unsure’ (n = 3), and ‘quite un-useful (n = 2). No owners felt that the questionnaire was ‘extremely un-useful. Sixteen people described the type or purpose of the questionnaire, categorised as overall well-being (6), registering or transferring to a new vet (5), treatment evaluation (2), arthritis (2), and pain assessment (1).

Ten (2.4%) owners reported having used a phone Application to complete questions about their dog’s health and well-being. Four owners found the App ‘extremely useful’, three ‘quite useful’, whilst two were ‘unsure’. Two people had used the PetDialog™ app, another used PetsApp, and the remaining were related to improving care, monitoring of a health condition (diabetes) before buying a puppy, a specific health check, and an online video vet service, with the remaining undisclosed.

Most owners (70.8%) reported that they would wish to access tools designed to assess various aspects of their dog’s behaviour, health, and well-being that they could complete in their own time, whilst 11.5% were unsure, and 17.7% said they would not.

### 3.4. Semi-Structured Interviews

Of those dog owners who participated in the survey and had reportedly completed a questionnaire with (or for) their vet about their dog’s health and well-being (n = 18), nine provided consent to be contacted about further related research. Of these, five were recruited, though one was subsequently unable to participate. Four interviews were completed and ranged between 20 and 41 min in length. The interviewees were all female; three were aged 35–44 years old, and one was between 55–64 years. Two worked in non-animal related professions, one was a dog trainer, and one was a researcher of animal behaviour. Participants described the purpose of the questionnaires they completed with their vets as follows: a broad health questionnaire when transferring to a new vet, a LOAD (Liverpool Osteoarthritis in Dogs) [17] questionnaire completed several times over four years, an emergency treatment questionnaire to establish medical histories and recent events, and general well-being (related to potential changes/development of arthritis).

Three core themes were generated during reflexive thematic analysis: ‘use of assessment tools supports client-vet relationship and empowers owners’, ‘owners want to talk about holistic dog care’, and ‘owner feelings on the wider application of assessment tools’. These themes are reported with illustrative quotes, within which all original meaning is maintained, but ‘[…]’ is used to shorten or contextualise where needed.

#### 3.4.1. Theme 1: ‘Use of Assessment Tools Promotes Client-Vet Relationship and Empowers Owners’

1.Promoting client-vet relationship

There was a wealth of feedback from interviewees suggesting that the use of structured questions aided communication and mutual understanding between them and their vet about their dog’s health and well-being, boosting the vet-client relationship and feelings of collaboration and being heard. One owner felt that working with their vet using the assessment tool helped them convey concerns and provided personal empowerment.


*“… just being believed by somebody today, and then you know about it being… a collaborative process. Just makes it kind of a lot easier because I have been in situations where I’ve been really sort of trying to convince the vet that I’m worried about something, and then they’ve been like, OK!” (Jenni and Bella)*


Using structured questions also gave owners a sense of trust that the veterinary team were providing a rigorous and rounded service.


*“It [the structured questionnaire] does provide comfort because, for me personally, It shows that they’re doing a thorough job. They need to know, you know, background and history and current events and what’s going on, to know what best to do next, to take the next best steps” (Katie and Trixie)*


It also appeared to help put veterinary questioning into context.


*“I’ve had many great vets down the years, but it can be nice to feel like you understand why they’re asking a question. As well as that, they’re asking something [and] it’s like okay, you’re actually looking for specific criteria here. [It] can be quite interesting to think, “Okay, why are you asking me that?” rather than just having a really super casual conversation.” (Rachel and Juno)*


Importantly, we read here how the use of structured questions made the owner feel that the vet was showing bespoke care for their dog:


*“I think it just made us really like her because it made us feel like she cared… It wasn’t just okay, bring your dog in, give them a shot, go home… I think the length of questioning made us feel more comfortable with her. Because we felt like okay, you’re really taking the time to get to know our dog, whereas we’ve had other vets who were a bit quick in the consultation” (Rachel and Juno)*


For clients, having the opportunity to complete assessments and to witness the making of additional records of their dog’s health and well-being appeared to help remove a sense of being rushed through the vet consultation and having to remember everything. Clients seemed to find confidence in how the vet took a structured interest in their dog’s life beyond the consultation room.


*“it all works pretty well and gives you a chance to be more reflective and more honest because I think in the vets, you’re like under pressure, you don’t remember everything properly and also you wanna try and like, like, please the vet as well.” (Jenni and Bella)*


We also get an impression from Jenni and Bella that owners can feel examined and pressured about articulating their dog’s health challenges through the consultation process. The structured recording of information appears to take the pressure off the client for needing to remember everything.


*“I think it just makes conversations at the vets just so much easier coz you don’t have to remember everything; they’ve got all the data they need, so you can just have a really productive conversation about… different treatments and things.” (Jenni and Bella)*


This interviewee also identifies a greater capacity for assessment and diagnosis through being asked for information that exceeds what could be picked up from examination and discussion during an isolated 15-min consultation.


*“I think it’s a good tool to have alongside whatever else you’re doing. Because in the balance, in that moment, to kind of assess some dogs in a 15-min consult[ation] for anything, if it’s something more complex and it’s really hard… then you’re able to see whether it’s just a real short-term change in quality of life, or is it something that started to go on like a bit more… long term?” (Jenni and Bella)*



*“it was through… those questionnaires that the vets could see… [arthritis] … spread to her [the dog’s] back and hips. And that’s the thing that’s impossible to sort of see in the vets” (Jenni and Bella)*


Another interviewee applauded the tool they experienced for accumulating a dog’s health records over a duration, which could give enhanced detail to vets when they assess an animal.


*“I think that… they’re a necessity, you know that the vets and the nurses… see hundreds of different animals and owners and for hundreds of different reasons they can’t be expected to retain all that knowledge themselves, although they’re great at you know, the relationship that it’s just not practical, so I think to have a record and keep asking, and you know, maintaining those records is crucial to doing the job well that they do.” (Katie and Trixie)*


2.Emotionally empowering and improved decision-making

The client-vet discussions that interviewees described revealed the intense emotional impact that caring for a dog with a chronic medical issue can have on owners and how challenging it can be for them to remain objective.


*“…when you look at it and go, she [the dog] had 22 bad days… but you always want a positive, don’t you, that hope. I think that’s what makes it difficult… You absolutely remember all the good bits and find it really, really hard to sort of objectively remember.” (Jenni and Bella)*


However, access to assessment tools that facilitated conversations with their vets over repeat appointments helped owners to record and measure deteriorations in their dog’s health over time and feel able to tackle the challenges and decision-making as things arose. Owners appeared to value the sense of control, support and reliable information gathering that the tools offered and the insight into changes in their dogs’ well-being over time. They described how it both helped them to understand the past and prepare for future dog care, expressing reassurance.


*“I think when you have those conversations and use that language early on, it changes your mindset, like completely. You think about everything in a totally different way, it makes you think more proactively and objectively, and it makes everything less emotional. Like, less about you… about them.” (Jenni and Bella)*



*“that [assessment tool] worked really well, so it kind of enabled you to kind of like predict what was sort of going to happen next, so they can see like sort of changes and then you know to try different things, and then be able to assess you know after that was actually working and helping” (Jenni and Bella)*



*“you’ve got a really strong metric to go back to and say, well, last time I said this, and now this score has changed from this to this, and it means we can actually track something rather than just trying to be like all, I guess, 2 years ago I said, maybe it was more like this it’s like no no you’ve got a record. You know what’s going on. You’ve got the data.” (Rachel and Juno)*


It is interesting to note how a structured assessment can support dog owners’ emotional well-being in their everyday anxieties towards whether they are caring well enough for their dogs.


*“I’m quite an anxious person who sort of doubts yourself and things like that. I just think it gives you that bit of support.” (Jenni and Bella)*



*“The positivity of like not just you care, but you’re on top of this, in a very organised way.” (Rachel and Juno)*


One owner reported that her and her vet’s use of structured questions over time aided decisions surrounding euthanasia and even helped it to become a positive experience after having had a very upsetting end-of-life experience with her previous dog, which was plagued with anxiety and self-doubt.


*“That’s [the tool] what helped me sort of make decisions… so our mantra, my vet says well it’s better 2 weeks early than 1 minute too late, and I had that in the back of my head.” (Jenni and Bella)*


The qualities of objectivity in an assessment tool are valued at a time when strong emotions can be experienced as euthanasia is contemplated.


*“I think especially, you know… towards the end, it does give you that realisation when you get, when you use those tools because it’s so objective you can’t argue with it. And when you can see, like, that their quality of life is perhaps, like, declining a little bit, or you know you’re getting to that sort of end-stage with your dog, like it’s quite emotional so… it’s really tough, because you can’t argue with sort of what’s there” (Jenni and Bella)*


The client was comforted about the timeliness of the decision-making to euthanise, which was assisted by using the tool.


*“When [dog] was euthanised, I think it was… the most positive experience it could have been. Because I knew that I literally did everything I could, and I knew that I couldn’t have done any more. And the vets, you’re having that conversation that it didn’t feel like a trauma of at the end it felt like we made that decision together… because it was about quality of life, but when we got to the end, for me personally, it made it a lot easier. Just there weren’t those feelings of guilt and shame. I didn’t have any of that this time, which is quite nice. It was as nice as it could have been.” (Jenni and Bella)*


#### 3.4.2. Theme 2: ‘Owners Want to Talk about Holistic Dog Care’

Interviewees were asked about conversations that they had had with their vets about quality of life, and these appeared to primarily occur when their dog’s physical health was compromised or in decline, much less if the dog was deemed to be in good physical health. This suggests an association with the application of tools and conversations with declining health and end-of-life decisions rather than ongoing welfare assessment in healthy animals.


*“I think it’s a conversation that we had right from when she was diagnosed with arthritis”. (Jenni and Bella)*



*“Not [spoken about QOL with vet] my current dog because, frankly, she’s fit and healthy” (Katie and Trixie)*



*“The main time they’ve done it [vet has discussed QOL], we took her in, and we said we think she’s starting to have a bit of arthritis” (Rachel and Juno)*



*“It is, in the end, the quality of life that declines, and so yeah, and you know, getting to that point that’s when it’s [QOL] been mentioned… with the vet, you know, we need to consider that now, instead of another method of managing the situation.” (Katie and Trixie)*


However, interviewees expressed a desire to discuss and consider their dog’s QOL earlier and throughout their lifetime, including access to tools that might inform health care and the opportunity to discuss issues beyond physical health.


*“I think, you know, initially, when people think about, or they hear about the quality of life, they do tend to associate that with the end of life things, but it should be from birth to the end of life because you want that quality of life throughout the lifespan of the animal, so if vets can help us achieve that, or working together, they can both achieve that, that’s probably the best thing to do.” (Katie and Trixie)*



*“I wish I’d had it [assessment tool] back when I had the dog with arthritis because maybe it would have picked up, that like an unwillingness to do that hill meant we should have been putting on pain meds much before we did because we just thought she was, she was always lazy… as she was getting older and it was exactly that thing of, actually it’s not age.” (Katie and Trixie)*



*“… vets tend to work in isolation. They think about the physical aspects, but they don’t consider all the others… although my current vet, the vet I have transferred to, is a fear-free practice, and she does understand how behaviour can affect physical health, the questionnaire that you complete is still quite basic and very health oriented… it’s so interlinked; the pain, behaviour, nutrition, it’s all linked together. I think saying what’s physically wrong with your dog or horse or cat is not providing the full picture.” (Jenni and Bella)*


It is important to acknowledge that client experience varied according to the scope of the questionnaire used, e.g., Mary and Barney, who completed a broad health questionnaire when transferring to a new veterinary practice, felt that the tool applied was not detailed enough.


*“it didn’t give me an opportunity to go into minute detail about the issues; I feel that we’d done that verbally” (Mary and Barney)*


#### 3.4.3. Theme 3: ‘Owners’ Feelings on the Wider Application of Assessment Tools’

1.Tools to educate and enhance conversations

Owners were asked for their thoughts on the wider application of structured assessment tools for different aspects of dogs’ lives between veterinary practitioners and dog owners. Questionnaires were seen as useful in improving owner education and understanding.


*“I think it should be proposed to each owner every time … I think it should be something that, as a matter of course, is asked… because in the long run, it’s only going to help serve the animal better and their treatment their care, you know, and it could even educate owners.” (Katie and Trixie)*



*“I think it would be very useful, yeah, especially because I’m trying to think about looking for signs in animals, and you know I’ve had a lot of dogs, I think, especially inexperienced owners, maybe don’t know what to look for or don’t know what to think about… you miss things because you don’t know what you’re missing.” (Rachel and Juno)*


Two interviewees conveyed a view that tools should enhance rather than replace conversational communication and decision-making.


*“I would see it very much more as something that you would complete and then preferably follow up with the vet, either over the phone or in person or over the phone or whatever. Erm, yeah, so I very much see the erm electronic communication just as a starting point before follow-up.” (Mary and Barney)*



*“I think the one thing to bear in mind; it doesn’t replace the relationship with the vet themselves or the vet’s own judgment because I know, having done research with people and with animals that, like, people also miss-report stuff” (Rachel and Juno)*



*“… at the end of the day, there is no substitute for talking to somebody.” (Mary and Barney)*


There was also a suggestion that tools may be intimidating and overwhelming to less knowledgeable owners, so a balance needs to be struck on the level at which questions are pitched to ensure clients will engage:


*“I think a tool that provides more details information would be more useful. Having said that… I have slightly better knowledge than most people, and I have a lot of awareness of what is going on and potentially going on with my dog than a lot of people. I would find that very useful. I think, potentially, some people might be slightly intimidated if they were asked to provide a lot of in-depth information or, or make a supposition about their dog.” (Mary and Barney)*


2.Preferred format of tools

Interviewees appeared to prefer electronic forms of assessment tools.


*“My personal preference would be electronic, not so much phone app. I find filling out things on phones really frustrating… some form of electronic or website would be the easiest way to do it these days.” (Mary and Barney)*



*“If we are trying to take that burden off, vets, with technology like if it was an app or even just something you could use more electronically… I think there’s a bit of a gap sometimes because the technology is quite old, so yeah, I’m, like, printing, and I’m, like, scanning it in or like sending an email. So, I think that it would be unbelievable if you have like an app or just something you can do electronically, like a form to fill in. It’d be so much easier, I think, for the vets to handle it… I think it might make owners more compliant or be a quick, easy thing to do.” (Jenni and Bella)*



*“Everybody’s using phone apps nowadays, aren’t they? They’re so accessible, you know, easy to update; they’re intuitive to use. So I think, probably following that format, it would be a really good one, you know, it’s got to be user-friendly, and everybody’s using phones, so it just seemed like the natural choice, to be fair. If each vet practice, whichever one you, you were signed up to, had like so you have like your own online dashboard with access to an app, say, you could update your pet’s detail…” (Katie and Trixie)*


However, there were differences in opinion about how regularly tool assessments should occur, which likely reflects varying owner experiences based on their dog’s current health status.


*“so at the end of the week, if you just quickly did this little app for some questions if it was attainable, I think there’s potential for something like that” (Jenni and Bella)*



*“… it could be good to check in like once a year… I wouldn’t want to do it too often, but I think it’s a really useful tool to look at, like is anything changing is anything that maybe you haven’t noticed you haven’t focused on… I wouldn’t want to have to do it every single vet visit or anything.” (Rachel and Juno)*


Interviewees varied in their preferences about accessing tools in their own time or at the veterinary practice and how they would see this working for other owners.


*“It’s better to do these things at home initially because I think one can feel slightly pressurised if you are filling something out with the vet, erm, and you don’t necessarily—erm—if you don’t know what the questions are you don’t have chance to think about them.” (Mary and Barney)*



*“When you’re in the vet, you want to, like not like intentionally, but you want to show them that you’re like trying and doing a good job. And I don’t think, even as humans and owners, we don’t always give like an honest reflection, like in the vets, you don’t always want to be open… I also think the two issues of kind of compliance, so, like, if you do it at home, a lot of people just forget about it and don’t do it. Or it all works pretty well and gives you the chance to be more reflective and more honest because I think in the vets, you’re like under pressure, you don’t remember everything properly and also you wanna try and like, like, please the vet as well.” (Jenni and Bella)*



*“I think you know as owners… if you take on any animal, you should do it [the tool]. Why, you know, why wouldn’t you? but, equally, you know, if people don’t have access to a phone or they don’t fill it in or if they don’t have the time, it can be revisited when the next time they do go into the vets.” (Katie and Trixie)*



*“I feel like if I get asked to do it at home, it will end up on being on my list of, like, things I should be doing that I don’t even remember to do. Maybe when you sit down at the vets at the beginning, and they, like, in that waiting period where you’re really just sat there looking at your phone or playing with the dog or trying to stop the dog eating the cat in the basket across the room. That could be a good time to just be like, okay, here’s a tablet, tick off these things, have a quick think, ‘cause that also puts you in the right frame of mind, then, to talk to the vet and he then came up with a question. You wouldn’t have to remember it for 3 or 4 days; you could just be like, okay, so I just saw on your survey that asked me about this and that made me think.” (Rachel and Juno)*


## 4. Discussion

This study described UK dog owners’ experiences and feelings about 17 conversation topics relating to the physical and emotional health of their dog with their veterinarian via feedback from an online survey. This was followed by a thematic analysis of owner interviews about their perceptions of tool-facilitated conversations about their dog’s health and well-being with their vets.

Survey respondents reported that veterinary professionals commonly discussed topics with them that related to physical health, with preventative medicine, weight and a specific health problem being the three most discussed topics. Topics relating to emotional well-being (e.g., behaviour problems, fear-related issues, mental stimulation, socialising) were discussed much more rarely. This may be because owners were attending appointments with their vets for specific health-related reasons as opposed to a wellness check of an apparently healthy dog, where it has been shown that veterinary communication differs depending on the focus of a consultation [18]. However, neglecting to address and explore various aspects of emotional health, such as fear-related issues or other behaviour problems, may compromise treatment efficacy and overall patient health and well-being. Unaddressed behaviour problems can lead to chronic stress [19], impact physiological and emotional health and welfare [20], contribute to a range of medical conditions, seriously challenge owner-dog relationships, impair learning, and in some cases, ultimately lead to abandonment, relinquishment, or euthanasia [19,21,22,23,24,25].

In the current study, most owners reported that they would be comfortable discussing behaviour problems and fear-related issues with their vets, as well as other topics relating to emotional well-being. They felt that these topics should be raised if the veterinary professional thought it was relevant, indicating expectations of a paternalistic model of communication in the veterinary environment. It should be emphasised here that owners appear to by relying on vets to raise such issues, and therefore, problems may not be detected or recognised without vets asking owners about them directly, and furthermore, this form of practitioner communication may inhibit client behaviour change [2]. The majority of owners felt that discussions about behaviour problems and fear-related issues should be addressed within a separate consultation, suggesting they would be willing to attend the vets specifically to discuss emotional health. Very few owners felt that these topics were not appropriate for discussion by veterinary professionals. Given up to 80% of the UK dog population is thought to experience some form of problematic behaviour [26], which poses a potentially considerable risk to canine welfare, vets may wish to consider the inclusion of more exploratory conversation around patient emotional well-being, and referral to paraprofessionals such as clinical animal behaviourists for treatment when problems are identified. Discussing other topics like puppy socialisation or training, mental stimulation, or socialising with other dogs or people more regularly may also provide opportunities to educate and therefore protect against the development of certain problematic behaviours through improving client awareness.

Euthanasia was the topic that most owners were least comfortable discussing, and the majority felt that it should be raised if the vet felt it was appropriate in a separate consultation. This is unsurprising, given the highly emotive and sensitive nature of such conversations, plus conversations about euthanasia are likely only raised when the end of life needs to be considered. It is important to note that feeling uncomfortable about discussing euthanasia does not necessarily equate to owners viewing it as not being a useful or important aspect of their dog’s life to discuss when needed. Disagreements can exist between owners and vets in reasons for considering euthanasia [27] as well as communication gaps between vets and clients in euthanasia consultations [4], so specific QOL assessment tools that have been recommended as part of a euthanasia ‘tool kit’ could be useful [28,29], where end-of-life decision making is a stressful decision for both vets [30,31] and owners [32]. Certainly, the current study suggests that the use of an assessment tool can indeed bridge communication gaps and reduce stress in this area. One interviewed dog owner expressed that the ongoing use of a tool not only assisted mutual decision-making between herself and her vet about the best timing for euthanasia but also provided emotional comfort that she was making the right decision “*it was… the most positive experience it could have been. Because I knew that I literally did everything I could, and I knew that I couldn’t have done any more*”.

Interestingly, the three most popular topics that owners said they would currently wish to discuss with their vet (a specific health problem, general welfare, and preventative medicine) had been reported by the majority as pre-existing discussion topics (although general welfare only held a marginal majority). From this, what owners most wanted to talk about with their vet appears to map well onto what had already been discussed. However, more people wanted to see issues such as behaviour problems and fear being addressed than had been experienced. Arguably, ‘welfare’ and ‘quality of life’ mean the same thing [33], but there may be individual differences in conceptualising QOL, where it is notoriously difficult to define [34], which might explain why fewer owners chose QOL as a priority topic to discuss with their vet compared with general welfare.

Most owners felt that their dog’s happiness and QOL should be addressed routinely in every veterinary consultation, yet only 32% reported that their vets had discussed QOL with them, suggesting an unmet demand where owners appear to expect to discuss their dog’s general well-being, irrespective of health status. A total of 95.8% of owners said they would be comfortable (77.5% very comfortable; 18.3% quite comfortable) discussing QOL. Whilst happiness and QOL may sound relatively ambiguous as conversation topics, in addressing this discrepancy, vets may be able to tap into rapport-building conversations about broader health and life-long QOL, which may reveal otherwise hidden issues, and could have beneficial impacts on patient management and treatment outcomes [18], including in healthy dogs [34]. The owners who completed the survey in this study had not previously used assessment tools designed to measure QOL. However, a high proportion (70.8%) expressed interest in accessing tools to assess different aspects of their dog’s behaviour, health and well-being that could be completed in their own time, which also strongly suggests that implicitly, owners are interested in discussing QOL and in learning more about their dog’s health and well-being. Certainly, within the interview analysis theme “owners want to talk about holistic dog care”, there is further support for the suggestion that owners would happily engage in conversations surrounding different aspects of their dog’s lives, covering topics that all essentially relate to QOL.

Of all the survey respondents, only 4.4% reported that they had completed a questionnaire with their vet about their dog’s health and well-being, which mirrors and adds further weight to the findings of Roberts 2022 [12], who found a low uptake of quality of life (QOL) assessment tools in veterinary practice. This suggests that, for the most part, vets are not using available assessment tools to aid, e.g., evaluation of treatment, screening for welfare issues, monitoring of disease, or decision-making about different aspects of canine health [35]. In the current study, none of the tools experienced by owners at their vets were QOL assessment tools but were questionnaires designed to assess a certain aspect of health (e.g., the LOAD questionnaire) [17] or more broad health questionnaires. Whilst some owners reported experience with tools that had received validity assessments, such as PetDialog™ [14] and LOAD [17], it is possible that there were other tools being used with limited scientific evaluation.

Within the theme “use of assessment tools promotes client-vet relationship and empowers owners,” it was demonstrated how interviewed dog owners felt that they were receiving more thorough, individualised care with a sense of collaboration and engagement between themselves and the vet. Assessment tools appeared to improve balance in the conversation between client and vet, moving away from a paternalistic model of communication and towards a shared, relationship-focused approach to care, giving owners the opportunity to learn and further engage with various aspects of their dog’s health. This may positively impact how clients respond to veterinary advice [2,3]. Suggestions for the application of assessment tools from interviewed owners include the use of electronic formats, being able to complete these in their own time at home or whilst in the waiting room, with the flexibility to engage repeatedly at varying time intervals. This is all encouraging feedback for veterinary professionals who may perceive owner resistance as a barrier to utilising QOL assessment tools [12]. Furthermore, this complements findings that showed clients who experienced a specific QOL tool felt that questions about their dog’s care were answered significantly more than those not using a tool [7], and 81% of clients felt more involved in their dog’s care after QOL tool use [8].

Vet visits all ultimately involve owner and vet engagement with an aspect of the dog’s health to either maintain or improve their QOL, yet vets and their clients do not appear to be engaging with ‘Quality of Life’ terminology. All the topics in the current study survey related to QOL, and most owners described being comfortable and wanting conversations about each topic, so implicitly, owners are interested in discussing QOL. It could be that researchers in this field, and veterinary professionals, need to consider bridging the gap surrounding the language and messaging used to make conversations about QOL more explicit, in addition to improving QOL assessment tool use. Future research may wish to investigate owners’ feelings on what QOL means to them to support the improvement of vet-client messaging.

One limitation for gaining a more in-depth understanding of owner perceptions in the current study was that very few owners who participated in the survey had used formal health or well-being tools with their vets, and of these, even fewer had given permission and provided details to be contacted about further research, so the interview population was small. Furthermore, the four owners who were interviewed experienced different tools. Having differences between the original purpose of the tool and only a small research sample means it is likely that data saturation within reflexive thematic analysis was not reached, warranting further study. On the other hand, interviewing owners who had completed diverse types of health and well-being questionnaires with their vets could be seen as accommodating and about exploring differences, which are key features of qualitative research. Future projects may wish to look at owners’ experience of different existing QOL tools used in practice (in addition to the study by Mwacalimba et al., 2020) [7]. A further limitation relates to the difficulty of understanding how representative the survey respondents’ responses were of the wider dog-owning population. Despite the wide posting of the survey within online dog-related fora, respondents who chose to participate may have had a particular interest in veterinary interactions or supporting science through completing research surveys than other dog owners, potentially making them more open to formal QOL assessments.

This study found that veterinary perceptions of owner-related barriers to discussing QOL and using QOL assessment tools [12] appear unfounded. There may be value in discovering and addressing other opinions that vets have about clients, which may impact client-vet relationships and the provision of patient care.

## 5. Conclusions

Whilst QOL remains hard to define and can cover every aspect of an animal’s life from birth to death, this study dispels one of the main barriers quoted by vets for the application of QOL assessment tools. Our results suggest that owners are not resistant to discussing topics relating to QOL and are open to using tools designed to assess health and well-being within a vet-led context. This offers veterinary professionals the opportunity to initiate conversations that may well have positive impacts on more holistic aspects of dog welfare, including in physically healthy animals. Considering how vets could improve these conversations, tool use may enhance the client-vet relationship and perceived quality of care rather than replace more implicit communication and assessment. Whilst peer-reviewed tools for assessing QOL are available yet remain largely unused in practice, veterinary professionals can be encouraged by our findings to use such tools with owners to improve the vet-client-patient relationship.

## Figures and Tables

**Figure 1 animals-13-00392-f001:**
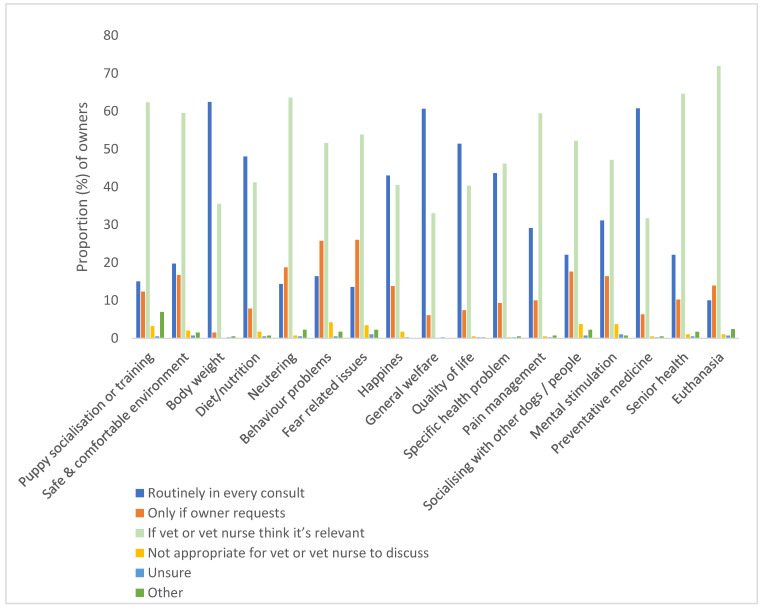
Proportion of owners (%) reporting when they felt a topic should be addressed by a vet or a vet nurse, if at all.

**Figure 2 animals-13-00392-f002:**
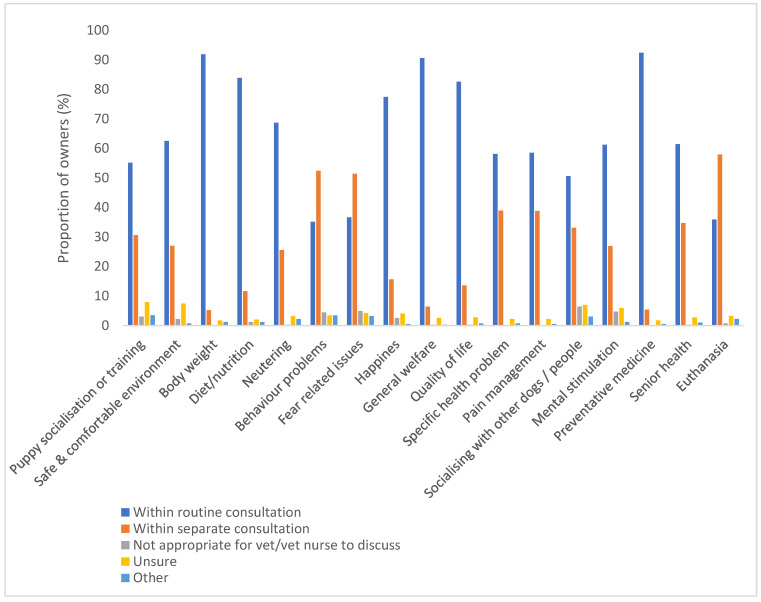
Proportion of owners (%) reporting how they felt each topic should be addressed at the vets, if at all.

**Figure 3 animals-13-00392-f003:**
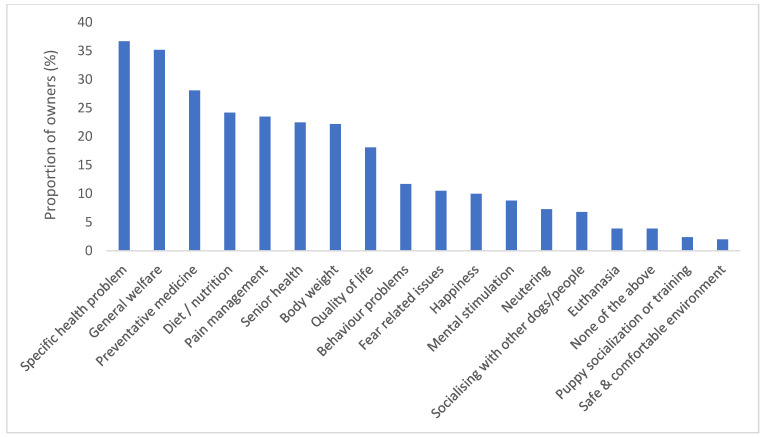
Proportion of owners (%) per topic that was preferred for current discussion with their vet or vet nurse (owners were asked to pick three topics).

**Table 1 animals-13-00392-t001:** Demographics of the dog owners who completed the online survey.

	Number of Owners (%)
**Gender**	
Male	29 (7.1)
Female	380 (92.9)
**Age category**	
18–24	22 (5.4)
25–34	81 (19.8)
35–44	87 (21.3)
45–54	87 (21.3)
55–64	92 (22.5)
65+	40 (9.8)
**Profession**	
Vet or vet nurse	14 (3.4)
Animal behaviourist	15 (3.7)
Dog trainer	15 (3.7)
Dog walker	15 (3.7)
Animal carer	9 (2.2)
Researcher of animal behaviour	10 (2.4)
Other animal-related profession	20 (4.9)
None of the above	311 (76)
**Number of dogs**	
1	264 (64.4)
2	107 (26.1)
3	20 (4.9)
4	6 (1.5)
5	3 (0.7)
6+	10 (2.4)
**Length of ownership**	
0–6 months	26 (6.3)
6–12 months	37 (9)
1–2 years	58 (14.1)
3–5 years	110 (26.8)
6–8 years	71 (17.3)
9–12 years	81 (19.8)
13+ years	27 (6.6)
**Previous ownership or first dog(s)?**	
First dog(s)	71 (17.3)
First dog(s) but grew up with a family dog	77 (18.8)
Previously owned a single dog	63 (15.4)
Previously owned >1 dog	199 (48.5)
**Number of annual vet visits**	
<1 year	25 (6.1)
1 year	100 (24.4)
2 year	133 (32.5)
3–4 year	106 (25.9)
5+ year	45 (11)

**Table 2 animals-13-00392-t002:** Number and proportion of owners reporting whether their vet or vet nurse had discussed each topic with them about their dog(s).

Discussion Topic	Yes N (%)	No N (%)	Unsure N (%)
Preventative medicine	370 (90.7)	34 (8.3)	4 (1.0)
Body weight	327 (80.1)	79 (19.4)	2 (0.5)
Specific health problem	308 (76.0)	94 (23.2)	3 (0.7)
Neutering	285 (70.9)	117 (29.1)	0 (0)
Diet/nutrition	248 (61.7)	153 (38.1)	1 (0.2)
General welfare	217 (54.5)	180 (45.2)	1 (0.3)
Pain management	196 (49.4)	196 (49.4)	5 (1.3)
Quality of life	129 (32.3)	266 (66.7)	4 (1.0)
Puppy socialisation or training	127 (31.9)	263 (66.1)	8 (2.0)
Senior health	89 (22.6)	297 (75.6)	7 (1.8)
Happiness	88 (22.4)	295 (75.1)	10 (2.5)
Behaviour problems	80 (20.1)	315 (78.9)	4 (1.0)
Fear related issues	79 (19.8)	317 (79.4)	3 (0.8)
Socialising with other dogs and/or people	76 (19.3)	312 (79.4)	5 (1.3)
Euthanasia	71 (18.0)	317 (80.5)	6 (1.5)
Creating a safe & comfortable environment	60 (15.2)	324 (81.8)	12 (3.0)
Mental stimulation	43 (11.0)	343 (87.7)	5 (1.3)

**Table 3 animals-13-00392-t003:** Number and proportion of owners reporting whether they felt comfortable discussing each topic with their vet about their dog(s).

Topic	Very Comfortable N (%)	Quite Comfortable N (%)	Neither Comfortable nor Uncomfortable N (%)	Quite Uncomfortable N (%)	Very Uncomfortable N (%)
Specific health problem	354 (86.8)	45 (11)	7 (1.7)	1 (0.2)	1 (0.2)
Preventative medicine	350 (85.4)	48 (11.7)	7 (1.7)	3 (0.7)	2 (0.5)
Pain management	340 (85.3)	53 (13)	12 (2.9)	1 (0.2)	1 (0.2)
Body weight	337 (82.4)	56 (13.7)	11 (2.7)	4 (1)	1 (0.2)
Neutering	324 (79.4)	55 (13.5)	18 (4.4)	7 (1.7)	4 (1)
Senior health	322 (78.9)	63 (15.4)	20 (4.9)	1 (0.2)	2 (0.5)
General welfare	321 (78.7)	73 (17.9)	11 (2.7)	2 (0.5)	1 (0.2)
Quality of life	317 (77.5)	75 (18.3)	11 (2.7)	5 (1.2)	1 (0.2)
Diet/nutrition	314 (76.8)	67 (16.4)	18 (4.4)	7 (1.7)	3 (0.7)
Creating a safe & comfortable environment	295 (72.3)	72 (17.6)	35 (8.6)	4 (1)	2 (0.5)
Happiness	295 (72.2)	68 (16.7)	32 (7.9)	9 (2.2)	2 (0.5)
Puppy socialisation or training	286 (69.9)	59 (14.4)	48 (11.7)	11 (2.7)	5 (1.2)
Mental stimulation	283 (69.4)	67 (16.4)	40 (9.8)	15 (3.7)	3 (0.7)
Socialising with other dogs/people	280 (68.8)	68 (16.7)	36 (8.8)	18 (4.4)	5 (1.2)
Behaviour problems	278 (68.3)	71 (17.4)	37 (9.1)	14 (3.4)	7 (1.7)
Fear related issues	278 (68.3)	70 (17.2)	39 (9.6)	13 (3.2)	7 (1.7)
Euthanasia	251 (61.8)	80 (19.7)	34 (8.4)	24 (5.9)	17 (4.2)

## Data Availability

The data presented in this study are available on request from the corresponding author.

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
