# Peer review of "Broadening the Veterinary Consultation: Dog Owners Want to Talk about More than Physical Health"

_animals, 2023, doi:10.3390/ani13030392_

Round 1

Reviewer 1 Report

Paper summary

This study aimed to improve understanding of UK dog owner comfort and use of conversations and tools for assessing their dog’s QOL. Using a mixed methods approach (interviews and online survey), authors show that nearly all of pet owners sampled were comfortably discussing dog QoL but less than a third reported their vets addressing the issue. Despite the majority of owners being open to formal assessment tools few reported using one. Interviews with a very small number of owners provides addition narratives of owner experience of QoL tools and conversations. Overall results show potential practical areas of improvement for veterinary practice and challenges existing perceptions.

General comments

The paper is well written and expands on the existing literature on the topic. The findings will be of great interest to the veterinary community and the dog owning public. Results have important dog welfare applications which can be applied by veterinary practitioners/canine professionals. One major limitation is the sample size n=4) for the semi-structured interviews. However, the limitation and its impacts on the result are clearly stated on in the discussion and theme reflect large quant survey findings.

Reviewer 2 Report

I think you could have gone a bit into the different metrics used and why they were used, which QOL scales are validated and which are not, but overall a good paper

Reviewer 3 Report

Check the attached file for comments. 
